# Disentangling the effects of intermittent faecal shedding and imperfect test sensitivity on the microscopy-based detection of gut parasites in stool samples

**Lana C. E. Ferreira-Sá**[1,2]**, Eleuza R. Machado**[2,3]**, Rodrigo Gurgel-Gonçalves**[1,2]*****, Fernando Abad-Franch**[1,4]

**1** Programa de Pós-Graduação em Medicina Tropical, Faculdade de Medicina, Universidade de Brasília, Brasília, Brazil, **2** Laboratório de Parasitologia Médica e Biologia de Vetores, Faculdade de Medicina, Universidade de Brasília, Brasília, Brazil, **3** Unidade de Farmácia e Dispensação Farmacêutica, Hospital Universitário de Brasília, Empresa Brasileira de Serviços Hospitalares/Universidade de Brasília, Brasília, Brazil, **4** Laboratório de Ecologia de Parasitos e Vetores, Departamento de Parasitologia, Instituto de Ciências Biológicas, Universidade Federal de Minas Gerais, Belo Horizonte, Brazil

\* gurgelrg@hotmail.com

**Data Availability Statement:** All relevant data are within the paper and its Supporting Information files.

## Abstract

### Background

Gut-parasite transmission often involves faecal shedding, and detecting parasites in stool samples remains the cornerstone of diagnosis. However, not all samples drawn from infected hosts contain parasites (because of intermittent shedding), and no test can detect the target parasites in 100% of parasite-bearing samples (because of imperfect sensitivity). Disentangling the effects of intermittent shedding and imperfect sensitivity on pathogen detection would help us better understand transmission dynamics, disease epidemiology, and diagnostic-test performance. Using paediatric *Giardia* infections as a case-study, here we illustrate a hierarchical-modelling approach to separately estimating the probabilities of host-level infection ($\Psi$); stool-sample–level shedding, given infection ($\theta$); and test-level detection, given infection and shedding ($p$).

### Methods/Findings

We collected 1–3 stool samples, in consecutive weeks, from 276 children. Samples (413 overall) were independently examined, *via* standard sedimentation/optical microscopy, by a senior parasitologist and a junior, trained student (826 tests overall). Using replicate test results and multilevel hierarchical models, we estimated per-sample *Giardia* shedding probability at $\hat{\theta} \approx 0.440 \pm 0.116$ SE and observer-specific test sensitivities at $\hat{p}_{\text{Senior}} \approx 0.639 \pm 0.080$ SE and $\hat{p}_{\text{Junior}} \approx 0.460 \pm 0.071$ SE. Gender-specific infection-frequency estimates were $\hat{\Psi}_{\text{Girls}} \approx 0.540 \pm 0.137$ SE and $\hat{\Psi}_{\text{Boys}} \approx 0.337 \pm 0.098$ SE. Had we used a (hypothetical) Perfect Test with 100% narrow-sense sensitivity ($p_{\text{PT}} \equiv 1.0$), the average probability of detecting *Giardia* in a sample drawn from an infected child ($\Psi \equiv 1.0$) would have been $\Pr(d|i, \text{PT}) = \Psi \times \theta \times p_{\text{PT}} \approx 1.0 \times 0.44 \times 1.0 \approx 0.44$. Because no test can be >100%

**Funding:** This work was supported by the Conselho Nacional de Desenvolvimento Científico e Tecnológico(314892/2021-4 to RG-G) and Coordenação de Aperfeiçoamento de Pessoal de Nível Superior (001 to LCE Ferreira-Sá). The funders had no role in study design, data collection and analysis, decision to publish, or preparation of the manuscript.

**Competing interests:** The authors have declared that no competing interests exist.

sensitive, $\Pr(d|i)$ (which measures *clinical* sensitivity) can only be brought above ~ 0.44 by tinkering with the availability of *Giardia* in stool samples (i.e., θ); for example, drawing-and-pooling 3 replicate samples would yield $\hat{\theta}_{3s} \approx 1 - (1 - 0.44)^3 \approx 0.82$.

## Conclusions

By allowing separate estimation (and modelling) of pathogen-shedding probabilities, the approach we illustrate provides a means to study pathogen transmission cycles and dynamics in unprecedented detail. Separate estimation (and modelling) of true test sensitivity, moreover, may cast new light on the performance of diagnostic tests and procedures, whether novel or routine-practice.

### Author summary

Gut parasites are transmitted through parasite shedding in the faeces of infected individuals. Particularly in resource-limited settings, diagnosis heavily relies on microscopy-based detection of parasites in stool samples. However, faecal parasite shedding is often intermittent, meaning not all samples drawn from infected hosts contain the parasite. Additionally, no diagnostic test has perfect sensitivity, and false-negative test results are common. To understand parasite transmission and diagnostic-test performance, we need to know both (1) how often a parasite is shed (shedding probability) and (2) what are the chances that a diagnostic test will detect the parasite when present in a sample (test sensitivity). Using *Giardia* infections as a case-study, here we illustrate a statistical-modelling approach that can be used to separately estimate parasite-shedding probability and diagnostic-test sensitivity. We ran duplicate microscopy-based tests on 413 stool samples drawn from 276 children, and found that *Giardia* shedding probability was ~44%; that single-test sensitivity was ~46–64%; and that, after accounting for intermittent parasite shedding and imperfect test sensitivity, infection frequency estimates (~34–54%) were over twice the observed frequencies (~16–25%). This general framework can help us better understand infection dynamics, epidemiology, and diagnosis whenever pathogen shedding is intermittent and pathogen detection is imperfect.

## Introduction

Intestinal infections remain a leading cause of global disease burden; while gut pathogens are common across world regions and human groups, they disproportionately affect children in underserved communities [1,2]. Intestinal parasites, including protozoa and helminths, substantially contribute to gut disease-related morbidity [3]. In routine practice, and notably in the resource-limited settings where intestinal-parasite infections are most relevant to public health, diagnosis involves using optical microscopy-based techniques to detect the parasites in stool samples [4,5]. Detection is vital not only for diagnosing individual-host infections, but also for understanding pathogen transmission dynamics and control, disease epidemiology and burden, and the effects of anti-parasitic treatment at the patient and population levels (e.g., [6]).

Despite their crucial importance and widespread use, all microscopy-based approaches to detecting parasites in stool samples have two key limitations. First, not all stool samples drawn

from infected hosts do contain parasites. This is due to the pervasive phenomenon of *intermittent parasite shedding*, which stems from, e.g., intermittent egg-laying by gut helminths or intermittent cyst formation by gut protozoans [4]. Shedding of infectious forms/stages is not only central to parasite transmission and epidemiology; in addition, when shedding is intermittent, some samples will lack parasites, giving rise to the paradox that even the most sensitive parasitological test will miss some infections [4]. Second, no test can detect the target parasites in 100% of the samples that indeed contain those parasites; this is due to the also pervasive phenomenon of *imperfect test sensitivity* [3,4,6]. Here, 'test sensitivity' is narrowly defined as the probability that a parasitological test detects the target parasite (in the form of eggs, larvae, cysts, trophozoites, or any other life-stage) in a sample in which that target is indeed present and, therefore, 'available' for detection. 'Target availability' is thus defined as the probability that a sample drawn from an infected host contains the detection target [7,8]. This rationale shows that the overall probability of detecting, by means of a given test, the presence of a target parasite in a stool sample drawn from an infected host is the product of two, possibly varying probabilities: (1) the probability that the target is available in the sample (which reflects shedding and we will call $\theta$), and (2) the probability that, when available, the target is detected by the test (which reflects narrow-sense test sensitivity and we will call $p$) (cf. [8]). This 'compound' probability of detection, given infection (which we write $\Pr(d|i) = \theta \times p$), is what clinicians see when testing a sample drawn from an infected host, and may hence be termed 'clinical sensitivity' (e.g., [9]).

The standard recommendation for improving clinical sensitivity is to draw replicate samples and then test them with single or replicate tests (e.g., [3,4,10]). If, for example, the average availability of a parasite in stool samples is $\theta \approx 0.50$ (i.e., the target is present in only about half of the samples drawn from infected hosts, due to intermittent shedding) and the average, narrow-sense sensitivity of a test of choice is $p = 0.75$, then running one test on one sample drawn from an infected host will yield a clinical sensitivity of $\Pr(d|i) \approx 0.50 \times 0.75 \approx 0.38$. Drawing three replicate samples would bring the probability that at least one of them contains the target to $\hat{\theta}_{3s} \approx 1 - (1 - 0.50)^3 \approx 0.88$, and blind duplicate testing of a target-containing sample would bring sensitivity to $\hat{p}_{2t} \approx 1 - (1 - 0.75)^2 \approx 0.94$. Thus, pooling three samples and running two tests on that pool would yield a clinical sensitivity of $\Pr(d|i, \hat{3s}, 2t) \approx 0.88 \times 0.94 \approx 0.83$ [11]. While this is a clear improvement, the single-number estimate of clinical sensitivity says very little about its two component probabilities; for example, knowing that $\Pr(d|i, \hat{3s}, 2t) \approx 0.83$ only tells us that $\hat{\theta}_{3s}$ and $\hat{p}_{2t}$ must lie somewhere between 0.83 and 1.0. Yet knowing $\theta$ would provide crucial insight into parasite shedding, which is a key driver of transmission, and knowing $p$ would provide crucial insight into test performance, which is key to interpreting the outcome of diagnostic procedures and to estimating epidemiological parameters including prevalence, incidence, or cure rates.

Here, we show how a multilevel hierarchical-modelling approach can be used to separately estimate $\theta$ and $p$, to model their variation, and to derive estimates of infection frequency that formally account for the vagaries of the sampling-testing process [7,8,10–12]. We illustrate our approach with a study of paediatric infection with the parasitic protozoan, *Giardia duodenalis* ('*Giardia*' hereafter). *Giardia* is a major global cause of diarrhoea, and transmission involves faecal shedding of environment-resistant infectious cysts that may then be ingested by a susceptible host [3,13,14]. Cyst formation, in turn, depends on several time- and space-varying gut-level factors (including lipid availability, bile-salt concentrations, or the aggregation of trophozoites in high-density foci) that combine to produce intermittent cyst shedding [13,15–18]. *Giardia* diagnosis heavily relies on the microscopy-based detection of parasites in stool samples [5]. Although there is consensus that intermittent parasite shedding and imperfect

test sensitivity can both impair clinical sensitivity [3,4,13,19], their individual magnitudes and variation remain unclear. To address this gap, here we (i) formally estimate the probabilities of parasite shedding (given infection) and microscopy-based detection (given shedding) associated with paediatric *Giardia* infections; and (ii) assess whether and to what extent (a) shedding probability varies with key host traits and (b) detection probability varies with observer expertise. The general framework we illustrate builds upon research on wildlife population ecology [12,20] and, we argue, can help us better understand infection dynamics, epidemiology, and diagnosis whenever pathogen shedding is intermittent and pathogen detection is imperfect [7,8,11].

## Materials and methods

### Ethics statement

The Institutional Review Board of the School of Medicine, University of Brasília, Brazil reviewed and approved this study (code 17596919.3.0000.5558). The parents or legal guardians of all children authorised the collection of stool samples and signed informed-consent forms.

### Study area and participants

We collected stool samples in eight public kindergartens in the Federal District, Brazil, between March and September 2019. The study kindergartens serve communities in four urban sectors with different degrees of overall social vulnerability—higher in the Midwest sector, intermediate in the West and South sectors, and lower in the Northeast sector (Table 1 and Fig 1) [21]. Social vulnerability is measured by 19 sector-level indicators of (i) urban infrastructure and environment; (ii) human capital and education; and (iii) income and employment [21]. There was no calculation for sample size; instead, all the children in each kindergarten were invited to participate in the study, and those whose guardians provided informed consent and stool samples were included in our analyses. Children participating in the study were between 4 and 73 months old at the time of sampling (mean 37.2 months, standard deviation [SD] 13.7); girls and boys were almost equally represented (Table 1). No child was under treatment for gut parasites or potentially associated conditions such as diarrhoea during or shortly before the sampling period.

**Table 1. Study areas and subjects, Federal District, Brazil, 2019.**

| Vulnerability[a] | Urban sector | Kindergarten | Gender | | Age (months) | | Total |
|---|---|---|---|---|---|---|---|
| | | | Girl | Boy | 4–36 | 37–73 | |
| Higher | Midwest | MW1 | 14 | 10 | 3 | 21 | 24 |
| | | MW2 | 16 | 13 | 6 | 23 | 29 |
| Intermediate | West | W1 | 20 | 17 | 23 | 14 | 37 |
| | | W2 | 8 | 22 | 11 | 19 | 30 |
| | | W3 | 23 | 19 | 34 | 8 | 42 |
| | South | S1 | 28 | 32 | 23 | 37 | 60 |
| | | S2 | 6 | 8 | 4 | 10 | 14 |
| Lower | Northeast | NE | 19 | 21 | 17 | 23 | 40 |
| Total | | | 134 | 142 | 121 | 155 | 276 |

[a] Overall level of social vulnerability [21]

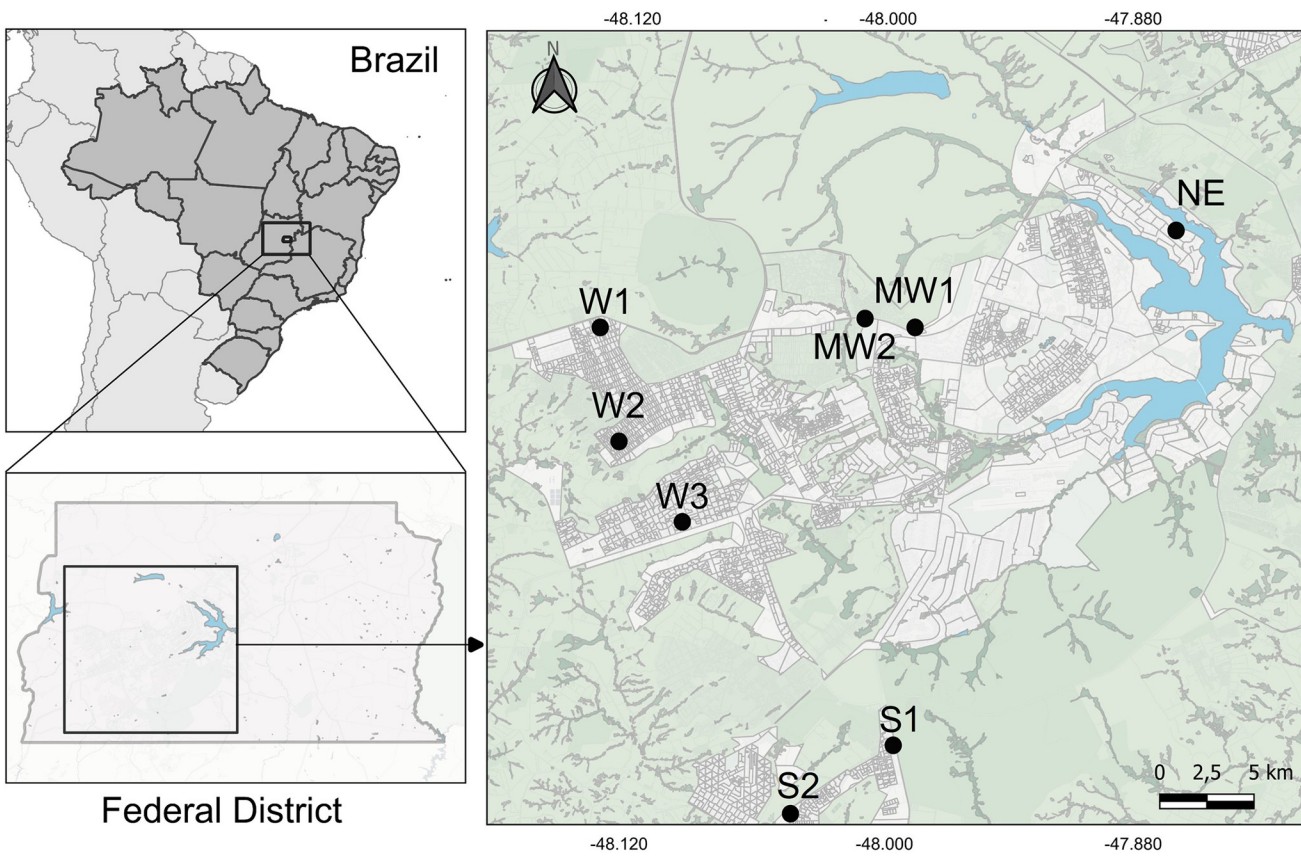

**Fig 1. Study area.** Location of the eight kindergartens participating in the study in four urban sectors of the Federal District, Brazil, 2019. The map was created using QGIS 3.36.1 (www.qgis.org) and publicly available shape files from the Instituto Brasileiro de Geografia e Estatística (IBGE; https://www.ibge.gov.br/geociencias/organizacao-do-territorio/malhas-territoriais.html).

## Collection, processing and examination of stool samples

We aimed to collect one stool sample per week over three consecutive weeks from each child. Samples were collected by child parents/guardians in 80-mL sterile, preservative-free collection bottles (J.Prolab, Paraná, Brazil). On Fridays, parents/guardians were given individually-labelled bottles together with oral and written instructions on how to collect and preserve stool samples, including the need for refrigeration. Collection bottles were retrieved on Mondays between 8 and 9 am, placed in Styrofoam boxes with four 200-mL reusable ice-gel packs (Gelo-Tech, Paraná, Brazil), and transported by car to the laboratory, where they typically arrived at around 10 am.

We processed our samples using the spontaneous sedimentation method—a simple, low-cost, low-technology method widely used in resource-limited settings (including in Brazil) to detect parasites in stool samples [22–28]. Briefly, we thoroughly mixed each sample with 50 mL of distilled water, filtered the resulting suspension through cotton medical gauze into a sedimentation cup, and washed the debris in the gauze by adding distilled water until the cup was filled. Sedimentation cups were then sealed and left standing for 24 hours at room temperature. We then carefully discarded the supernatant, homogenised the sediment pellet in 150 mL of distilled water, and left the samples at rest for 24 additional hours. After again discarding the supernatant, we homogenised the sediment in 10 mL of distilled water and transferred the

mix to 15-mL, sterile Falcon tubes; these samples were stored at 4˚ C until retrieval for slide preparation and reading.

Two independent observers (a senior parasitologist and a trained, yet novice graduate student) separately prepared and read three microscopy slides from each sample. Slides were prepared with one drop of thoroughly homogenised sample mix, stained with lugol, and read using Olympus BX41 optical microscopes (Olympus Corporation, Tokyo, Japan) at 400× magnification; when in doubt, specific structures were observed at 1000×. The entire field delimited by the coverslip (24×40 mm) was examined, with no time limit. Each observer prepared and read their slides without knowing either (i) the results seen by the other observer using slides from the same sample or (ii) the results of testing previous samples from the same child. We note, however, that observers knew which three-slide sets they had prepared from each sample; same-sample results provided by a particular observer, therefore, cannot be considered independent, and we did not analyse them separately [29].

### The data: coding and structure

Our data describe whether or not (coded "1" and "0", respectively) each observer detected at least one *Giardia* cyst or trophozoite in at least one of the three slides from each sample. Detections were only considered as such (and coded "1") when *Giardia* identification was unambiguous; we thus excluded potentially false-positive results, effectively trading sensitivity (which we can estimate) for specificity (which we will assume is ≈ 100% [30]). For each child in the dataset, we therefore have a *Giardia* 'detection history' with up to six binary (0/1) entries per stool sample (Fig 2). In what follows, "test" is defined as the reading, by one observer, of the three slides prepared with a given sample. For each child, the results of either two (just one stool sample provided), four (two samples provided), or six tests (three samples provided) are thus available for analysis; missing sample-specific observations from children who provided just one or two samples were coded "-". Note that the structure of the data (see S1 Dataset) reflects a three-level hierarchy in which tests are nested within samples nested within children (Fig 2). By including replication at each level of the hierarchy, our sampling-testing protocol can provide insight into children-level infection; stool-sample–level parasite shedding (or 'availability'), given infection; and test-level detection, given shedding [8,20]; see also, e.g., [31].

### Covariates

We aimed at illustrating how our approach can be used not only to estimate the probabilities of *Giardia* infection, shedding, and detection, but also to assess whether and how those probabilities vary with key, selected covariates. We considered the possibilities outlined below.

1. Child-level infection ($\Psi$) may be more likely in city sectors with overall higher social vulnerability (covariate 'vulnerability'; three-level factor, dummy-coded), and perhaps among older children with longer exposure times (covariate 'age'; months, continuous, standardised to mean 0 and SD 1); given the age range of our study participants, we expected little, if any, differences in infection probabilities between girls and boys (covariate 'gender'; binary indicator), perhaps with slightly higher values among males (e.g., [32–35]).

2. Stool-sample–level shedding, given infection ($\theta$), might be more likely among older children if *Giardia* gut-population densities tended to increase with time since infection, thus inducing steadier cyst formation and elimination [15]; we did not expect shedding to vary with child gender or among time-ordered serial samples (first/second/third; 'order' covariate, dummy-coded).

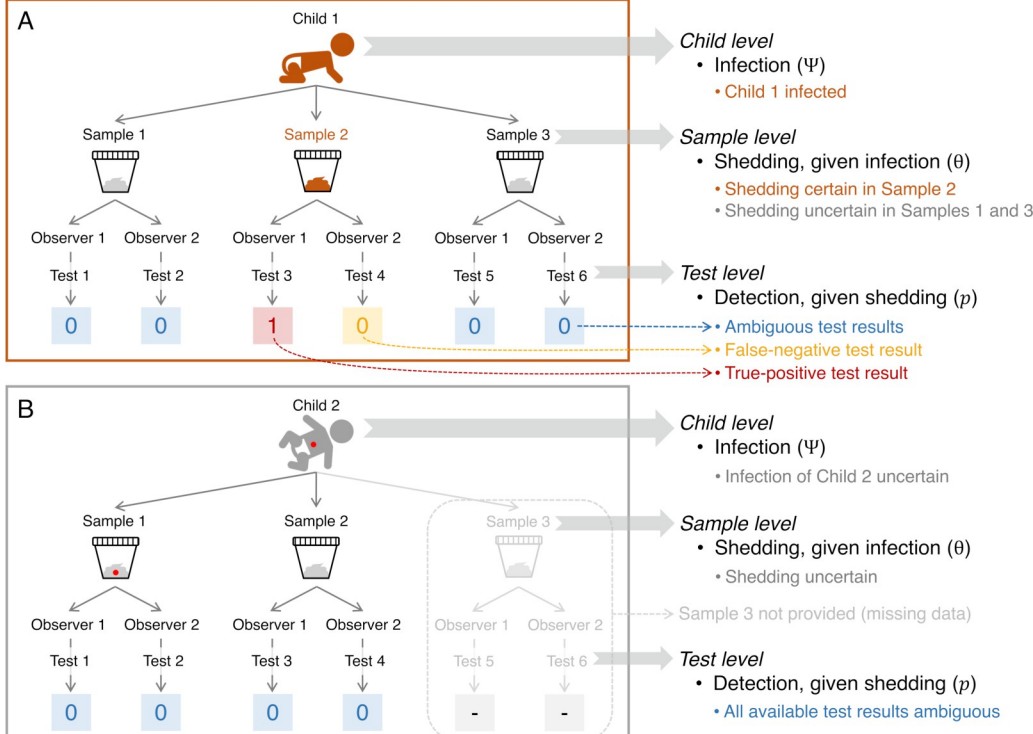

**Fig 2. Detecting *Giardia* in children: sampling-testing strategy and outcome coding.** Each child provided up to three stool samples, and each sample was tested twice (once by each of two independent observers). The result of each test (the reading of three microscope slides) was coded as "1" when at least one *Giardia* cyst or trophozoite was unambiguously identified in at least one slide, and "0" otherwise; when a child did not provide a sample, the results of the missing tests were coded "-". We depict two hypothetical examples. In panel A, Child 1 provided three serial samples (Samples 1 to 3). *Giardia* was detected in one of the tests run on Sample 2; this true-positive result shows that the child was infected and that there was shedding in Sample 2, although one test (by Observer 2) failed to detect the parasites (a false-negative result). None of the four tests run on Samples 1 and 3 yielded detections; for each of those samples, this can be due to either (i) no parasite shedding or (ii) false-negative test results. Therefore, the results of the four tests run on Samples 1 and 3 are ambiguous (highlighted in blue), and whether there was parasite shedding in those two samples remains uncertain. In panel B, Child 2 provided just two samples (Samples 1 and 2), and all four test results were ambiguous; child-level infection and sample-level shedding are therefore uncertain. Also in panel B, the red dots in Child 2 and Sample 1 highlight the possibility that infection is present and shedding occurs—but the parasites then go undetected by the (imperfect) tests.

3. Test-level detection, given shedding ($p$), may be more likely when a senior parasitologist reads the slides than when a less experienced, junior observer does the reading ('observer' covariate; two-intercept coding, with one intercept for the senior observer and another for the junior observer). In addition, test sensitivity might increase with child age if (assuming that age correlates with time since infection) older infections resulted in not only steadier (see point 2 above), but also more profuse cyst formation—thus leading to higher cyst densities within positive stool samples.

## Data analyses

We first described our data and did a series of exploratory analyses to preliminarily assess the evidence, if any, for (i) intermittent *Giardia* shedding, (ii) imperfect test sensitivity, and (iii) variation in observed child-level infection, sample-level shedding, and test-level detection. The hypothesis of intermittent faecal shedding predicts that not all samples drawn from children known to be infected will contain *Giardia*, and the hypothesis of imperfect test sensitivity

predicts that not all tests run on samples known to contain *Giardia* will yield *Giardia* detections. Note, however, that if both tests fail to detect the parasites in some *Giardia*-containing samples, then (a) the fraction of positive samples among those drawn from children known to be infected will often misestimate shedding, and (b) the fraction of positive tests among those run on samples known to contain *Giardia* will overestimate test sensitivity. If all the tests run on the samples drawn from some infected children come up negative (because of shedding absence or complete detection failure), then child-level infection frequency will be underestimated [12]. To better understand the patterns and drivers of *Giardia* infection, we therefore need to move beyond crude, biased observations and try to formally estimate (and model) three key parameters—host-level infection, sample-level shedding, and test-level sensitivity. We do this by adapting the multilevel modelling approach developed by Nichols et al. [20] to study wildlife ecology when detection of the target species is imperfect.

Multilevel site-occupancy models consider a hierarchy of (1) sites or habitat patches (i.e., spatial units), which may be occupied (with probability $\Psi$) by the target species; (2) sampling stations or occasions, in/during which the target species, if present at the spatial unit, may be available for detection (with probability $\theta$); and (3) observers, who may detect the target species, when it is available for detection, with probability $p$ [20]. In our application, *Giardia* is the target species; children are discrete 'spatial units' that *Giardia* may 'occupy' (i.e., infect) with probability $\Psi$; individual stool samples are analogous to 'sampling stations' in which *Giardia* may be available for detection with probability $\theta$ (which is conditional on $\Psi$); and individual tests are analogous to 'observers' and detect *Giardia* with probability $p$ (which is conditional on $\theta$) (cf. [8]). The probabilities of infection ($\Psi$), shedding ($\theta$), and detection ($p$) can all be modeled as a function of covariates using a (logistic) generalised linear modelling framework [11,12,20] (see also [7]).

We fit 38 different models (S1 Table), each representing a specific hypothesis about infection, shedding, and detection. For example, the hypothesis that *Giardia* infection frequency varied across genders can be represented by the model $\Psi(\text{gender}), \theta(.), p(.)$, in which the 'gender' covariate on $\Psi$ indicates whether each child was a girl or a boy (see Table 1). The "(.)" parts of this model denote that shedding and sensitivity are hypothesised to be constant; letting test sensitivity $p$ vary with observer expertise (in the expectation that the test will be more sensitive in the hands of more experienced observers) leads to the alternative model $\Psi(\text{gender}), \theta(.), p(\text{observer})$. We also considered an alternative model in which $\theta$ was fixed at 1.0; this specification represents the (worth testing but hardly plausible) hypothesis that shedding is 'perfect'—when a child is infected, *all* stool samples drawn from that child will contain *Giardia* (i.e., constant shedding), and any failure to detect the parasites will be due to imperfect (narrow-sense) test sensitivity.

With all alternative models fit to the same data, Akaike's information criterion (AIC) and related metrics can be used to measure the relative support that each model (and its associated hypothesis) finds in those data [36]. To do this, we fit our models *via* maximum likelihood in the free software PRESENCE 2.13.39 [37], and computed sample size-corrected AIC scores (AICc), AICc differences between each model and the top-performing model ($\Delta$AICc), and Akaike weights ($w_i$, a normalised measure of the relative likelihood of each model, given the data); smaller AICc and larger $w_i$ values indicate better model performance and, therefore, more empirical support for the associated hypothesis [36]. In a second inferential step, we look at the numerical output of (i) the better-performing models, including the top (smallest-AICc) model, and (ii) the entire model set, *via* model averaging [36]. For ease of interpretation, we focus on model-predicted probabilities (of infection, shedding, and detection) and their standard errors; in the case of model-averaged predictions, we account for model-selection uncertainty by computing unconditional standard errors [36].

## Results

### *Giardia* detection at the child, sample, and test levels

We ran 826 tests (three-slide readings) on 413 samples collected from 276 children (Table 2). The observed frequency of child-level infection was ~20%, with somewhat higher average values in the most vulnerable city sector, among girls *vs.* boys, and, to a lesser extent, among older *vs.* younger children (Table 2). Only ~69% of the 91 stool samples drawn from children known to be infected yielded detections, with apparently small differences associated with child age or gender. While observed variation among time-ordered serial samples might suggest a decline in positivity (from ~75% for the first sample to ~50% for the third), there is substantial uncertainty about sample-specific results (Table 2). Finally, only ~68% of the 126 tests run on

**Table 2.** *Giardia* detection in 826 microscopy-based tests run on 413 stool samples drawn from 276 children in eight kindergartens of the Federal District, Brazil, 2019.

| Level | Covariate[a] | Value | *n* | Positive[b] | Proportion ± SE[c] |
|---|---|---|---|---|---|
| Child | | | | | |
| Overall | - | - | 276 | 56 | 0.203 ± 0.024 |
| | Vulnerability | Higher | 53 | 15 | 0.283 ± 0.062 |
| | | Intermediate | 183 | 32 | 0.175 ± 0.028 |
| | | Lower | 40 | 9 | 0.225 ± 0.066 |
| | Child age | 4–36 | 121 | 23 | 0.190 ± 0.036 |
| | | 37–73 | 155 | 33 | 0.213 ± 0.033 |
| | Child gender | Girl | 134 | 34 | 0.254 ± 0.038 |
| | | Boy | 142 | 22 | 0.155 ± 0.030 |
| Sample | | | | | |
| Overall | - | - | 413 | 63 | 0.153 ± 0.018 |
| From positive children[d] | - | - | 91 | 63 | 0.692 ± 0.048 |
| | Sample order | First | 56 | 42 | 0.750 ± 0.058 |
| | | Second | 27 | 17 | 0.630 ± 0.093 |
| | | Third | 8 | 4 | 0.500 ± 0.177 |
| | Child age | 4–36 | 41 | 27 | 0.659 ± 0.074 |
| | | 37–73 | 50 | 36 | 0.720 ± 0.063 |
| | Child gender | Girl | 55 | 37 | 0.673 ± 0.063 |
| | | Boy | 36 | 26 | 0.722 ± 0.075 |
| Test[e] | | | | | |
| Overall | - | - | 826 | 86 | 0.104 ± 0.011 |
| On positive samples[f] | - | - | 126 | 86 | 0.683 ± 0.041 |
| | Observer | Senior | 63 | 50 | 0.794 ± 0.051 |
| | | Junior | 63 | 36 | 0.571 ± 0.062 |
| | Child age | 4–36 | 54 | 38 | 0.704 ± 0.062 |
| | | 37–73 | 72 | 48 | 0.667 ± 0.056 |

[a] Covariates measure (i) contextual (city sector-level) social vulnerability, (ii) child age in months (with children grouped, for illustrative purposes, in two strata—younger *vs.* older than approximately 3 years), (iii) child gender, (iv) stool sample order, and (v) observer expertise

[b] Unambiguous detection of at least one *Giardia* cyst or trophozoite

[c] Proportion positive ± approximate standard error (SE), computed as $\sqrt{(\pi(1-\pi))/n}$, where $\pi$ is the sample proportion and $n$ is the number of observations [38]

[d] Stool samples drawn from children with at least one positive test (i.e., known to be infected)

[e] Each 'test' consisted of the reading of three microscope slides by one observer

[f] Tests run on stool samples with at least one positive test (i.e., known to contain *Giardia*).

samples known to contain *Giardia* yielded detections, with a somewhat lower value for the junior observer and little variation associated with child age (Table 2).

While these results suggest that the probability of stool-sample–level shedding, given infection, was likely < 1.0, and indicate that the probability of test-level detection, given shedding, was indeed < 1.0, they nevertheless do not provide any bases for estimating those probabilities with confidence, for modelling their variation as a function of covariates, or for assessing the relative support that different hypotheses about infection, shedding, and detection find in the data.

## Multilevel site-occupancy modelling

We fit 38 models—each representing a specific hypothesis about child-level *Giardia* infection ($\Psi$), stool-sample–level *Giardia* shedding ($\theta$), and test-level *Giardia* detection ($p$)—to our dataset (see S1 Table for the full model set and S1 Dataset for the raw data). The 'null' model M0, $\Psi(.), \theta(.), p(.)$, estimates (i) average child-level infection frequency at $\hat{\Psi}_{M0} \approx 0.443 \pm 0.110$ SE, which is much larger and more uncertain than the observed frequency ($0.203 \pm 0.024$ SE; see Table 2); (ii) average sample-level shedding, given infection, at $\hat{\theta}_{M0} \approx 0.445 \pm 0.120$ SE; and (iii) average test-level detection, given shedding, at $\hat{p}_{M0} \approx 0.535 \pm 0.065$ SE. While these average estimates are a clear improvement over intuition derived directly from raw observations (Table 2), we may gain further insight by assessing the support for the 'null' hypothesis, relative to alternative hypotheses. For example, a more realistic, 'nearly-null' hypothesis might state that test sensitivity should vary with observer expertise; more specifically, sensitivity should increase when optical microscopy-based tests are run by senior observers. Model M1, $\Psi(.), \theta(.), p(\text{observer})$, represents a simple version of this hypothesis. Model M1's AICc score is 487.80, which is 2.95 units smaller than that of the 'null' model M0 (AICc 490.75; S1 Table); this difference provides some evidence against the 'null' hypothesis that test sensitivity is about the same for tests run by senior and junior observers. Model M1 instead suggests that observer seniority improved test sensitivity from $\hat{p}_{M1-\text{Junior}} \approx 0.460 \pm 0.071$ SE to $\hat{p}_{M1-\text{Senior}} \approx 0.639 \pm 0.081$ SE. Based on this result, and on the overall plausibility of observer effects, we included the 'observer' covariate in all our models except the 'null'; we also tested for observer effects by removing the 'observer' covariate from the top-performing model (see below) and checking for changes in AICc.

A comparison involving the full 38-model set (S1 Table) shows the following.

1. The top-performing model is $\Psi(\text{gender}), \theta(.), p(\text{observer})$, which represents the hypothesis that infection varied with child gender, shedding was constant (relative to the covariates we evaluated), and test sensitivity varied with observer expertise.

2. Apart from the top model, nine models have substantial empirical support ($\Delta$AICc < 2.0) and eight additional models have some support too ($\Delta$AICc 2.0–4.0) [36]. These 18 'competitive' models (shown in Table 3) suggest that (i) gender likely modulated child-level infection risk to some extent, with age and (contextual) social vulnerability perhaps contributing also to variation; (ii) sample-level shedding was probably similar across genders but may have varied slightly with child age; and (iii) test-level sensitivity varied with observer expertise (see also below), but not with child age.

3. A version of the top-ranking model in which we fixed sample-level shedding at $\theta \equiv 1.0$ had essentially no empirical support ($\Delta$AICc 15.66; $w_i = 0.00$; Tables 3 and S1); this refutes the (*a priori* implausible) 'perfect shedding hypothesis' and provides compelling evidence for intermittent faecal shedding.

**Table 3. Modelling *Giardia* child-level infection, sample-level shedding, and test-level sensitivity: top-ranking (and lowest-ranking) multilevel site-occupancy models.** Shaded cells indicate inclusion of the covariate in the infection (black shade), shedding (dark-grey shade), and/or sensitivity (light-grey shade) parts of each model. See S1 Table for the full model set.

| Rank | Infection (Ψ) | | | Shedding (θ) | | Sensitivity (p) | | AICc | ΔAICc | $w_i$ | k | Deviance |
|---|---|---|---|---|---|---|---|---|---|---|---|---|
| | Gender | Age | Vulnerability | Gender | Age | Observer[a] | Age | | | | | |
| 1 (top) | ■ | | | | | ░ | | 485.95 | **0** | 0.119 | 5 | 475.73 |
| 2 | ■ | | | | ▩ | ░ | | 486.34 | **0.39** | 0.098 | 6 | 474.03 |
| 3 | ■ | ■ | | | | ░ | | 486.66 | **0.71** | 0.083 | 6 | 474.35 |
| 4 | ■ | | ■ | | | ░ | | 486.71 | **0.76** | 0.081 | 7 | 472.29 |
| 5 | | | | | ▩ | ░ | | 487.78 | **1.83** | 0.048 | 5 | 477.56 |
| 6 | | | | | | ░ | | 487.80 | **1.85** | 0.047 | 4 | 479.65 |
| 7 | | | ■ | | | ░ | | 487.80 | **1.85** | 0.047 | 6 | 475.49 |
| 8 | | | | ▩ | | ░ | | 487.86 | **1.91** | 0.046 | 5 | 477.64 |
| 9 | | ■ | | | | ░ | | 487.86 | **1.91** | 0.046 | 5 | 477.64 |
| 10 | ■ | | | | | ░ | | 487.90 | **1.95** | 0.045 | 6 | 475.59 |
| 11 | ■ | | ■ | | ▩ | ░ | | 488.20 | 2.25 | 0.039 | 8 | 471.66 |
| 12 | ■ | ■ | | | ▩ | ░ | | 488.45 | 2.50 | 0.034 | 7 | 474.03 |
| 13 | ■ | ■ | ■ | | | ░ | | 488.49 | 2.54 | 0.033 | 8 | 471.95 |
| 14 | ■ | | | | | ░ | | 488.88 | 2.93 | 0.028 | 4 | 480.73 |
| 15 | | | ■ | | ▩ | ░ | | 489.36 | 3.41 | 0.022 | 7 | 474.94 |
| 16 | | ■ | ■ | | | ░ | | 489.43 | 3.48 | 0.021 | 7 | 475.01 |
| 17 | ■ | ■ | ■ | | ▩ | ░ | | 489.74 | 3.79 | 0.018 | 9 | 471.06 |
| 18 | | ■ | | | ▩ | ░ | | 489.86 | 3.91 | 0.017 | 6 | 477.55 |
| 38 (lowest) | ■ | | | Fixed (1.0) | | ░ | | 501.61 | 15.66 | 0.000 | 4 | 491.39 |

AICc, sample size-corrected Akaike's information criterion; ΔAICc, difference in AICc between each model and the top-ranking model (**bold** values highlight models with ΔAICc < 2.0 and, hence, with substantial empirical support); $w_i$, Akaike weights; k, number of estimable parameters; Deviance, twice the negative log-likelihood of each model; Gender, a binary covariate indexing child gender (female/male); Age, a standardised covariate describing child age, in months; Vulnerability, a three-level factor measuring contextual (city sector-level) social vulnerability (high/intermediate/low); Observer, a binary covariate indexing observer expertise in parasitology (two-intercept coding)

[a] The 'observer' covariate was included (*via* two-intercept coding) in all models except the 'null' model; to further check for observer effects, we re-ran the top-ranking model after removing this covariate, leading to the 14th-ranking model shown here (ΔAICc 2.93, $w_i$ 0.028)

4. Finally, variations of the top-performing model in which we (i) removed observer effects and (ii) tested for child-age effects on *p* had, respectively, modest (ΔAICc 2.93; Table 3) and little (ΔAICc 4.92; S1 Table) empirical support.

To provide a quantitative view of modelling results, we next examine numerical model output, focusing on probability-scale predictions for ease of interpretation (see S1 Text for the untransformed coefficient estimates and SEs from the 10 top-ranking models shown in Table 3). We start with the predictions of the top-ranking model, Mtop, which estimates (i) gender-specific child infection frequencies at $\hat{\Psi}_{\text{Mtop-Girls}} \approx 0.540 \pm 0.137$ SE and $\hat{\Psi}_{\text{Mtop-Boys}} \approx 0.337 \pm 0.098$ SE; (ii) the probability of stool-sample–level *Giardia* shedding, given child infection, at $\hat{\theta}_{\text{Mtop}} \approx 0.440 \pm 0.116$ SE; and (iii) observer-specific test sensitivities at $\hat{p}_{\text{Mtop-Senior}} \approx 0.639 \pm 0.080$ SE for the senior observer and at $\hat{p}_{\text{Mtop-Junior}} \approx 0.460 \pm 0.071$ SE for the junior observer (Fig 3).

Because several candidate models had substantial empirical support (Table 3), we next computed model-averaged predictions and unconditional SEs for each probability of interest, including sample-specific shedding. We note that PRESENCE 2.13.39 provides model-averaged

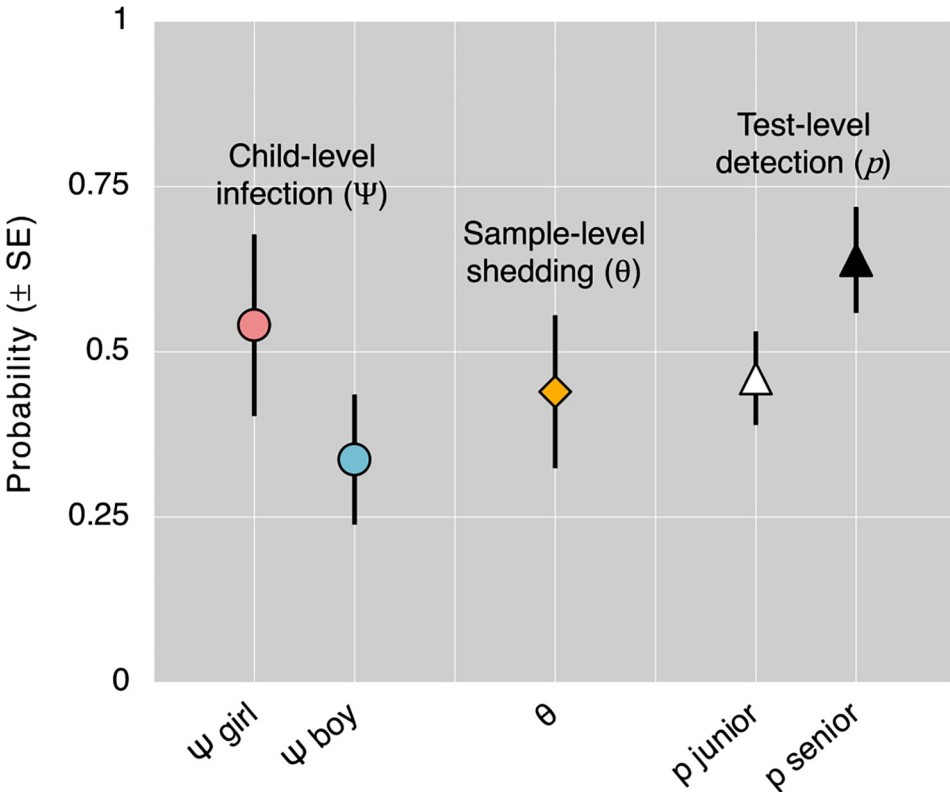

**Fig 3. Predictions from the top-ranking multilevel site-occupancy model: probabilities of child-level *Giardia*** infection (**Ψ**); sample-level *Giardia* shedding, given infection (**θ**); and diagnostic test-level *Giardia* detection, given shedding (i.e., narrow-sense test sensitivity, *p*).

predictions only for observed (in-sample) combinations of covariate values; in Fig 4 we show selected examples—namely, the highest and lowest estimates of child-level infection for each gender and social-vulnerability stratum; sample-level shedding for each gender; and test-level sensitivity for each observer. These results illustrate (i) how *Giardia*-infection frequency was moderately higher among girls than among boys, varied little and inconsistently with (contextual) social vulnerability, and perhaps increased slightly with age (Fig 4A); of note, Fig 4A also shows how naïve indices derived from raw observations grossly underestimate both infection frequencies and their associated uncertainties; (ii) how *Giardia* shedding was similar across serial stool samples and across genders, but appeared to increase moderately with age (Fig 4B); and (iii) how the sensitivity of our simple, low-cost, microscopy-based *Giardia* diagnosis test clearly increased with observer expertise, but did not vary with child age (Fig 4C).

## Discussion

Pathogen transmission depends on the ability of infectious forms or life-stages (mastigotes, zoites, cysts, eggs, larvae, bacterial cells, virions, etc.) to leave an infected host and then reach and colonise another, susceptible host. This process often involves *shedding*—the release of pathogens in the faeces or bodily fluids (respiratory droplets or aerosols, saliva, urine, etc.) of an infected host (Fig 5). When mediated by blood-feeding arthropod vectors (mosquitoes, flies, bugs, ticks, etc.), transmission depends on a special form of 'shedding'—the *availability* of pathogens within the bloodmeal taken by the vector (Fig 5). Importantly, host infection guarantees neither shedding nor availability; both will occur with probability < 1.0, so that not

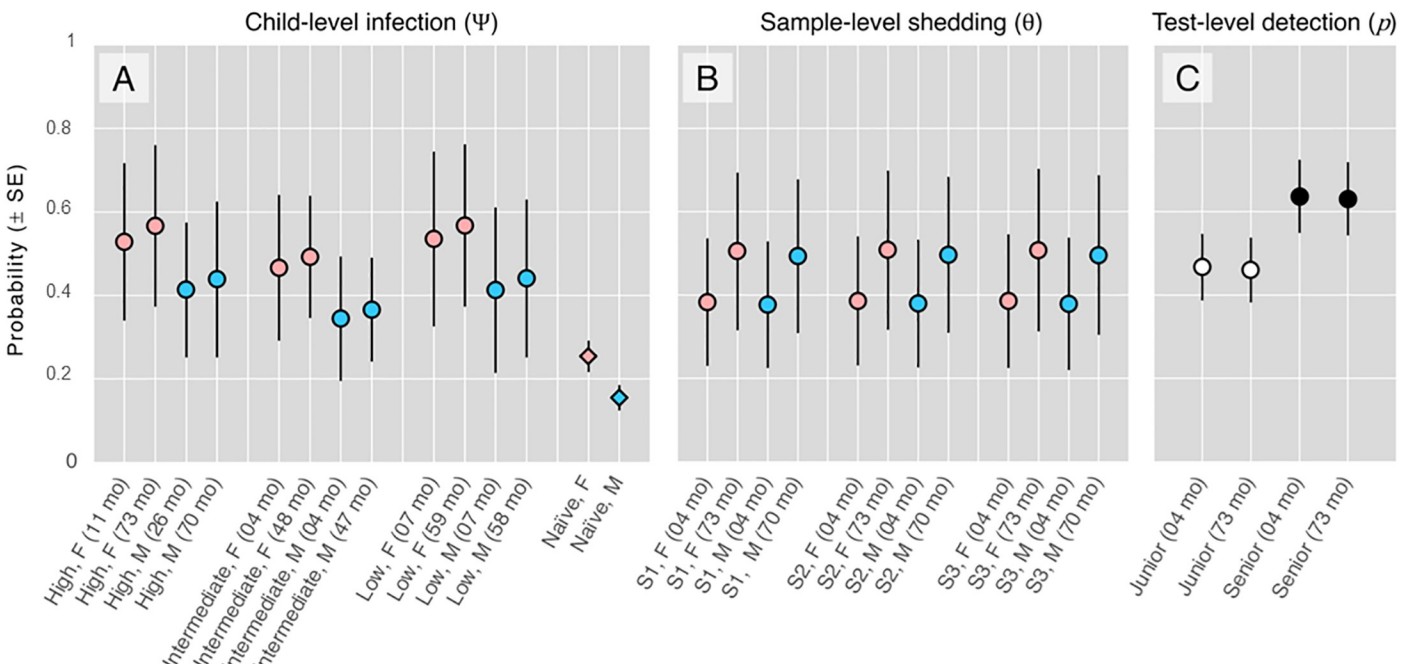

**Fig 4. Model-averaged predictions and unconditional standard errors (SEs).** A, model-averaged probabilities of child-level *Giardia* infection; naïve gender-specific frequencies (and approximate SEs; see Table 2) are also plotted for comparison. B, model-averaged probabilities of stool-sample–level *Giardia* shedding, given infection. C, model-averaged probabilities of test-level *Giardia* detection, given shedding (i.e., narrow-sense test sensitivity).

every stool, droplet, sputum. . . or blood sample including vector bloodmeals will contain the pathogen. This phenomenon of 'imperfect shedding/availability' affects not only pathogen transmission, but also infection diagnosis—even the most sensitive test cannot detect its target pathogen (or surrogate biomarker) in a sample that does not contain it. While critical to understand pathogen transmission and infection diagnosis, formal estimates of the probabilities of pathogen shedding/availability, given host infection, and of diagnostic-test performance, given shedding, remain largely unavailable [7–9,11,31]. Using paediatric *Giardia* infection as a case-study, here we have illustrated how an approach that combines robust sampling-testing designs with multilevel site-occupancy models can be adapted to (i) estimate those two key parameters (along with the probability of host infection) and (ii) model them as a function of covariates.

As expected, we found clear evidence of intermittent *Giardia* shedding in the stool of infected children [15,19,39,40]. We highlight two key findings. First, our analyses show that, relative to models allowing for intermittent shedding, a fixed-$\theta$ model representing the hypothesis of 'perfect' shedding ($\theta \equiv 1.0$) had no support from the data (Table 3). Second, our top-model and model-averaged estimates suggest that only about $44.0 \pm 11.6\%$ SE (range, ~38–51%; see Figs 3 and 4) of stool samples drawn from infected children did contain *Giardia*. Crucially, these statistical estimates of the (per-sample) probability of shedding, given infection, formally take imperfect test sensitivity into account—and our results show how far from 100% was the narrow-sense sensitivity of a standard, three-slide microscopy-based parasitological test. Specifically, our best estimates of sensitivity ($63.9 \pm 8.0\%$ SE for the senior observer and just $46.0 \pm 7.1\%$ SE for the junior observer) suggest that *Giardia* likely went undetected in about half (~46–64%) of the stool samples that not only came from infected children, but also contained *Giardia*. These probability estimates can be combined to derive an

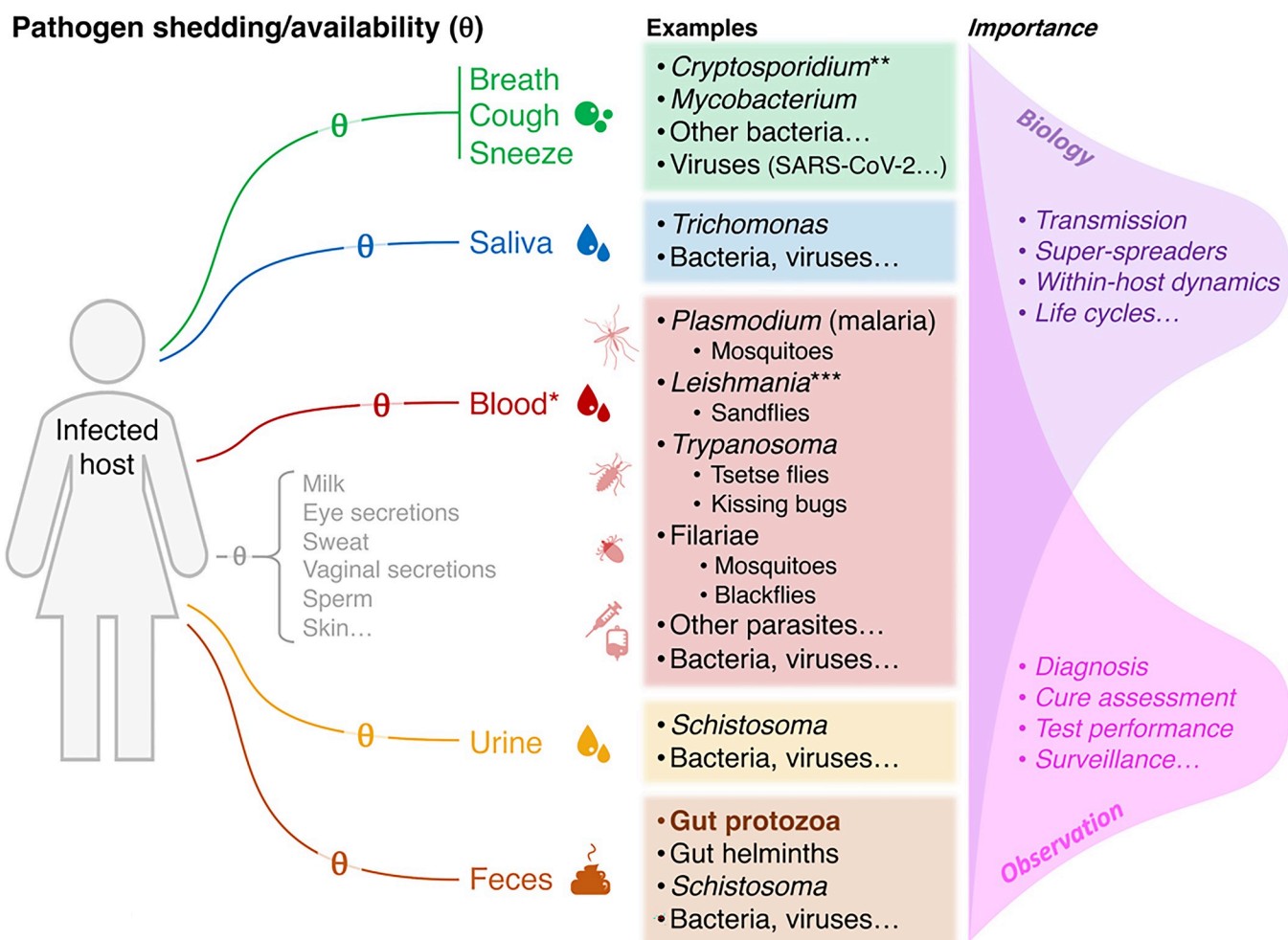

**Fig 5. Pathogen shedding/availability (θ): routes, examples, and importance.** The diagram stresses the complexity of pathogen shedding (or availability), the diversity of major pathogens depending on shedding for transmission, and the importance of measuring shedding to develop a rigorous understanding of both (i) the biological underpinnings of pathogen transmission cycles and dynamics and (ii) the observation process underpinning pathogen diagnosis and surveillance. The case-study described in this report (*Giardia*) belongs in the gut protozoa, and is highlighted in bold typeface. *Pathogens may be 'available' (with probability θ) in either vector bloodmeals or blood samples drawn from infected hosts. **For respiratory transmission of *Cryptosporidium*, see, e.g., [42]. ***Female sandflies pick *Leishmania* parasites when they ingest infected dermal macrophages together with a bloodmeal (see, e.g., [43]).

approximation to the probability of detecting infection in a host that is indeed infected (i.e., with $\Psi \equiv 1.0$), which we write $\Pr(d|i) = \Psi \times \theta \times p$ and measures 'clinical' sensitivity. If a single stool sample is used to run a single (three-slide reading) test, then (dropping $\Psi$, which equals 1.0) $\Pr(d|i, \text{Senior}) \approx 0.440 \times 0.639 \approx 0.281$ for our senior observer and $\Pr(d|i, \text{Junior}) \approx 0.440 \times 0.460 \approx 0.202$ for our junior observer. Hence, the joint consequence of intermittent parasite shedding and imperfect test sensitivity is that most *Giardia* infections (perhaps ~70–80% of them, as in our study setting) are likely to go undetected when a one-sample/one-test strategy is used for diagnosis. Previous studies in which two-level hierarchical models were used to estimate the 'clinical sensitivity' (i.e., $\theta \times p$) of microscopy-based *Giardia* diagnosis reported values ranging from ~0.14 to ~0.58 [28,41]; while our clinical sensitivity estimates (~0.20–0.28) fall within this wide range, we stress that differences in methods, settings, and populations (e.g., testing procedures, observers, infection intensity and clinical correlates) complicate the direct comparison of numerical results.

How can clinical sensitivity be enhanced? Using tests with higher narrow-sense sensitivity (i.e., higher $p$) is perhaps the most intuitive option. Imagine, for example, a highly sensitive test (called 'HST') that combines a highly efficient cyst-concentration step with a high-affinity fluorescent antibody and thus detects *Giardia*, when present in the sample, with average $p \approx 0.95$ 5. If one such HST were run on a single sample drawn from one of our infected children, then $\Pr(d|\hat{i}, \mathrm{HST}) \approx 1.0 \times 0.440 \times 0.955 \approx 0.420$, and 'just' ~58% of infections would be missed. It is then possible to further increase overall $p$ by running independent test replicates; with 2 HST replicates, $\hat{p}_{2\mathrm{HST}} \approx 1 - (1 - 0.955)^2 \approx 0.998$, and $\Pr(d|\hat{i}, 2\mathrm{HST}) \approx 1.0 \times 0.440 \times 0.998 \approx 0.439$. Importantly, this exercise shows that even a (hypothetical) *perfect* test ('PT') with perfect narrow-sense sensitivity ($p_{\mathrm{PT}} \equiv 1.0$) would only bring clinical sensitivity to ~44%, because $\Pr(d|\hat{i}, \mathrm{PT}) \approx 1.0 \times 0.44 \times 1.0 \approx 0.44$; to get clinical sensitivity beyond that upper bound, we therefore need to increase $\theta$ [8]. This can be done by collecting and pooling (with extra care to ensure thorough homogenization) multiple samples drawn within a short time-frame (to ensure that infection status does not change over the sampling period). Using three such samples, $\hat{\theta}_{3s} \approx 1 - (1 - 0.44)^3 \approx 0.824$, and blind duplicate HST-testing on those pooled samples would yield a clinical sensitivity of $\Pr(d|\hat{i}, 3s, 2\mathrm{HST}) \approx 1.0 \times 0.824 \times 0.998 \approx 0.822$. Another strategy to increase $\theta$ is to identify biomarkers of parasite presence that are more widely (bio-)available at the sample level (i.e., are shed at higher frequencies) than the parasites themselves; this might include parasite-derived molecules, such as antigens or nucleic acids, or host-derived molecules, such as antibodies [4,8]. The approach we illustrate here can help estimate the probabilities of sample-level shedding, given host-level infection, for these and other potentially useful biomarkers.

Formal estimates of shedding probabilities can not only help us better understand the process of diagnosing infections; when focused on *infectious* life-stages, including *Giardia* cysts and many others (Fig 5), they can also provide insight into pathogen transmission dynamics [7]. For example, the ability to separately estimate and model shedding probabilities opens the possibility of asking crucial questions about the underpinnings of among-host variance in infectivity—including why, for many pathogens, only a few individual hosts are 'super-spreaders' that account for most transmission events (e.g., [44–46]). By modelling shedding/availability of infectious life-stages as a function of covariates describing host characteristics, we can outline a 'super-spreader host profile' as the combination of covariates that maximise $\theta$ in a given host-pathogen system. In our case-study, for example, we found some evidence that older children may shed *Giardia* cysts at a higher frequency than younger ones; gender, however, was a poor predictor of shedding (Table 3, Figs 3 and 4). Because inducing host diarrhoea may be adaptive for parasites that depend on faecal shedding for transmission, we also predicted that shedding would be more likely in diarrhoeic samples. Our data, however, did not allow us to test this hypothesis (just six children had diarrhoea; see S1 Dataset), but our approach can, in principle, be used to estimate the effects of diarrhoea (or other clinically relevant variables) on shedding probabilities.

To our knowledge, this study is the first to provide formal statistical estimates of shedding probabilities for a human enteric pathogen (see [7,11] for applications to prions and trypanosomes). It has, however, some limitations that should be kept in mind when interpreting its results. First, the analyses are based on a rather small sample size, leading to relatively large uncertainties (SEs) around many of our estimates (Figs 3 and 4). Uncertainty about shedding probabilities was further amplified by the fact that only 113 children provided 2 stool samples, and just 24 provided 3 samples; while our models accommodate missing observations (by skipping them in the computation of the likelihood; see [12]), denser sampling would have yielded more precise $\theta$ estimates. Similarly, narrow-sense test sensitivity $p$ could have been estimated

with extra precision by blinding observers to the origin of each of the slides and then analysing the results of each slide-reading separately. In contrast, the uncertainty associated with our infection-frequency estimates should probably be larger than we report; this is because our models assume child independence with respect to infection status [20], yet infections in children sharing kindergarten and/or family space are likely to be correlated to a certain degree. Finally, our models also assume $\approx$ 100% test specificity—i.e., that no positive result was a false positive [20]. Even though the assumption seems realistic [30], we took extra care to only score a slide as positive when identification of *Giardia* was unambiguous. We close this paragraph on limitations by noting that, in any case, the main message of this report is not about the specific values of child-level $\Psi$, sample-level $\theta$, or test-level $p$ in our study system, but rather about how those key parameters can effectively be estimated while formally taking the vagaries of the sampling-and-testing process into account—an approach that may find application in a wide variety of pathogen–host systems (e.g., Fig 5 and refs. [7,8,11]).

In conclusion, here we have shown how separate estimation (and modelling) of pathogen shedding probabilities ($\theta$) is feasible and can potentially lead to a deeper understanding of pathogen transmission cycles and dynamics. Separate estimation (and modelling) of true, narrow-sense test sensitivity ($p$), moreover, may cast new light on the performance of diagnostic tests and procedures, whether novel or routine-practice. Equipped with such knowledge about the sampling-detection process, one can then (i) design diagnostic algorithms (i.e., sampling/testing strategies) that maximise clinical sensitivity, $\Pr(d|i)$, and (ii) derive more reliable estimates of infection frequency ($\Psi$), which are needed to better understand disease epidemiology and burden and also provide the basis for the planning, deployment, and rigorous evaluation of control-surveillance strategies or therapeutic interventions.

## Supporting information

**S1 Dataset. Raw *Giardia* detection dataset.**
(XLSX)

**S1 Table. The full set of 38 multilevel occupancy models fit to the *Giardia* dataset.**
(PDF)

**S1 Text. Untransformed coefficient estimates and SEs from the 10 top-ranking models in Table 3.**
(TXT)

## Acknowledgments

We thank Nadjar Nitz, Mariana M. Hecht, and Ciro M. Gomes for their suggestions. Josania Santos and Patricia G. de Assis helped with sample collection and processing.

## Author Contributions

**Conceptualization:** Rodrigo Gurgel-Gonçalves, Fernando Abad-Franch.

**Data curation:** Lana C. E. Ferreira-Sá, Eleuza R. Machado, Rodrigo Gurgel-Gonçalves, Fernando Abad-Franch.

**Formal analysis:** Lana C. E. Ferreira-Sá, Rodrigo Gurgel-Gonçalves, Fernando Abad-Franch.

**Funding acquisition:** Rodrigo Gurgel-Gonçalves.

**Investigation:** Lana C. E. Ferreira-Sá, Eleuza R. Machado.

**Methodology:** Eleuza R. Machado, Fernando Abad-Franch.

**Project administration:** Rodrigo Gurgel-Gonçalves.

**Resources:** Eleuza R. Machado, Rodrigo Gurgel-Gonçalves.

**Supervision:** Rodrigo Gurgel-Gonçalves.

**Validation:** Fernando Abad-Franch.

**Writing – original draft:** Lana C. E. Ferreira-Sá, Rodrigo Gurgel-Gonçalves, Fernando Abad-Franch.

**Writing – review & editing:** Eleuza R. Machado, Fernando Abad-Franch.

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
