## [Decision Letter · Decision Letter 0]

25 Sep 2024

Dear Dr. Gurgel-Gonçalves,

Thank you very much for submitting your manuscript "Disentangling the effects of intermittent faecal shedding and imperfect test sensitivity on the microscopy-based detection of gut parasites in stool samples" for consideration at PLOS Neglected Tropical Diseases. As with all papers reviewed by the journal, your manuscript was reviewed by members of the editorial board and by several independent reviewers. In light of the reviews (below this email), we would like to invite the resubmission of a significantly-revised version that takes into account the reviewers' comments. 

We cannot make any decision about publication until we have seen the revised manuscript and your response to the reviewers' comments. Your revised manuscript is also likely to be sent to reviewers for further evaluation.

Sincerely,

Alberto Novaes Ramos Jr

Academic Editor

Guilherme Werneck

Section Editor

Reviewer's Responses to Questions

**Key Review Criteria Required for Acceptance?**

**Methods**

-Are the objectives of the study clearly articulated with a clear testable hypothesis stated?

-Is the study design appropriate to address the stated objectives?

-Is the population clearly described and appropriate for the hypothesis being tested?

-Is the sample size sufficient to ensure adequate power to address the hypothesis being tested?

-Were correct statistical analysis used to support conclusions?

-Are there concerns about ethical or regulatory requirements being met?

Reviewer #1: yes

Reviewer #2: The methods are well described. The authors take a very technical approach to answering a very technical questions. Nevertheless I agree with the authors that their manuscript will be of interest to the NTD community, and is applicable to a lot of NTDs. I am afraid that I no familiar with the multilevel ecological modeling approach they've taken, and cannot confidently comment on its appropriateness. However it seems sounds to my uneducated understanding. I may have missed this, but it would be good to understand how the authors dealt with missing samples. It look they had 276 children over three week, which is 828 samples. However, there were only 413 samples in the results. How was missinginess accounted for, and how might affect the results? It is very understandable that sample collection approach (this age group, and done by parents) would result in missing samples, but it would just be good to think about how this affects the results.

Reviewer #3: The authors apply a hierarchical modeling approach to estimate Giardia infection in children attending kindergartens in the Federal District, Brazil while accounting (estimating) for stool-sample–level shedding rates and imperfect test-level detection. This type of approach is appropriate, but rarely applied, to disentangling the effects of intermittent shedding and imperfect sensitivity and better understand transmission dynamics, disease epidemiology, and diagnostic-test performance. We applaud the authors for applying such an approach and provide several suggestions to make the use of this approach more appealing to the journal’s audience and disease ecology in general. Below, we highlight several general suggestions, but also include smaller, editorial suggestions in the attached pdf.

1. The authors may consider enhancing the biological justification for their hypotheses (i.e., models). This could include a little more background on Giardia – it is a gut parasite, but how is it likely transmitted among individuals (i.e., children) in this system. The authors give some background on lines 113-127, but it is not clear how differentiating intermittent parasite shedding and imperfect test sensitivity helps manage, contain, or control pediatric Giardia infections. How does this enhanced understanding lead to better treatment for such diseases in this community, or similar communities? 

2. This understanding is particularly important to appreciate some of the models, and model covariates, investigated in this study. In theory, the covariates (lines 209-229) relate to hypotheses the authors want to test with their data but understanding the reasoning for some of these hypotheses (covariates) is not clear. For example:

a. The authors state that children were sampled from kindergartens in four urban sectors of the Federal District, Brazil with different degrees of overall ‘social vulnerability’. What criteria were used to determine ‘social vulnerability’? What makes children in one kindergarten more ‘socially vulnerable’ to pediatric Giardia infections than other communities/kindergartens?

b. The authors state that the probability that a child has Giardia infections may be higher with older children (age = months) or that older children may have higher intermittent shedding rates (probabilities), given infection (θ), because Giardia gut-population densities tended to increase with time since infection. The authors state that age is standardized (mean 0, sd = 1) but its not clear that they treated age as a continuous covariate. Table 2 suggests that the age were grouped into two categories: children 4-36 months; children 37-73 months. What is the significance of this grouping with respect to a child having Giardia infections or shedding environment-resistant infectious cysts in their stool? If the authors used age as a continuous variable, then I would suggest not giving data in age categories (e.g. Table 2) and reporting and graphing the relationship between a given parameter (e.g., theta, ϑ) and age over the range of values in their data (4-73 months), using a supported model (Model 2: ψ(gender) ϑ(age) p(observer); model weight =0.39). Similarly, the authors could have reported the slight effect of age on a child having Giardia infections (ψ), using a supported model (Model 3: ψ(gender+age) ϑ(.) p(observer); model weight =0.71).

c. The authors could provide some justification for why they considered time-ordered serial samples as a covariate if they did not expect this to influence shedding probability. 

3. Tables 1 and 2 seem a little redundant to me. Is Table 1 necessary? Alternatively, it might be possible to remove Table 2 or move to supplemental information since some of the highlights are given in this paragraph. Another option would be to retain Table 2 and add model-based estimates for these same levels.

4. Data Analysis Section: 

a. We are not sure explaining the multilevel occupancy models in terms of its historic use (i.e., with patches of habitat) is that useful to this audience. It might be a distraction. Instead, the authors could use the term 'units' instead of 'sites'. MacKenzie et al. 2017 chose to do so because of work like this that points out multi-level occupancy can be applied to units other than habitat patches. If the authors adopt this terminology, then the sentence on lines 248-250 would read:

“Multilevel site-occupancy models consider a hierarchy of (1) units (e.g., individuals), which may be occupied (with probability Ψ) by the target parasite/pathogen species; (2) sub-units or occasions, in/during which the target species, if present at the unit, may be available for detection, and...”

Additional changes may be needed later, if the authors adopt this terminology.

b. In addition to the comments regarding the authors hypotheses and associated covariates, the authors should provide more detail regarding their model building and model selection approach. Specifically:

i. Provide more detail on ‘the series of exploratory analyses to preliminarily assess the

 evidence, if any, for (i) intermittent Giardia shedding, (ii) imperfect test sensitivity, and (iii) variation in observed child-level infection, sample-level shedding, and test-level detection’. How did this lead to your candidate model set?

ii. I would like to see a bit more description in the supporting material for the model building and model selection process. For example, there is something called age_1 and age_2, which appears different from 'age'.

iii. Some of the models that they mention in the text (e.g., ψ(vulnerability) ϑ(.) p(.)) are not part of this model set.

iv. The authors might consult Morin et al. 2020 for a model building strategy:

Morin, D.J., Yackulic, C.B., Diffendorfer, J.E., Lesmeister, D.B., Nielsen, C.K., Reid, J. and Schauber, E.M., 2020. Is your ad hoc model selection strategy affecting your multimodel inference?. Ecosphere, 11(1), p.e02997.

 Their candidate model set is not balanced, and I question if that influences some of their inference is simply because of the lack of balance. For example, it’s not surprising that there was no effect of age on detection probability if only one model contained that structure, and the remaining 37 models did not. Similarly, it’s not surprising that there was no effect of gender on shedding probability if only one model contained that structure, and the remaining 37 models did not. Again, I would refer the authors to Morin et al. 2020 for more defendable, step-wise approach to model building and model selection.

Reviewer #4: This section is well-detailed and explains the processes sufficiently to ensure reproducibility. It's also fascinating to see how a model originally used in wildlife ecological studies is being applied to medical research. However, two points stand out:

While the primary focus is elsewhere, the sampling process in the kindergartens could be described in more detail. For example, were all children in the kindergartens included, or just a subset? How was the sampling conducted, and was there any calculation for sample size? Were there children who opted out of the study? Were any of the children under treatment for conditions such as diarrhea?

While it makes theoretical sense to exclude ambiguous reports to assume 100% specificity, it is unclear whether these ambiguous results were more common in the junior observer’s reports. If this were the case, excluding them could potentially skew the observer covariate calculations. Reporting the number (or at least the approximate percentage) of excluded cases could also be informative.

**Results**

-Does the analysis presented match the analysis plan?

-Are the results clearly and completely presented?

-Are the figures (Tables, Images) of sufficient quality for clarity?

Reviewer #1: yes

Reviewer #2: Yes, the results are good. Again, I am not an expert in this modeling approach, but it is relatively well presented.

Reviewer #3: The writing and communication of the results could be improved – see specific comments in the attached document. I do not think it is necessary to repeat the information included in the model selection Table 3 (Lines 338-354); instead, the authors should provide a better biological interpretation of these results. First, I would verify that your model selection table is balanced - i.e., verify that gender, age, and vulnerability appear in the same number of child-level Giardia infection (Ψ) structures. Similarly, verify that age, gender, order, and null (.) structures appear in the same number of stool-level Giardia infection (ϑ) structures. Again, see Morin et al. 2020 on various step-wise model building strategies that insure a balanced model set. 

In this candidate set, model 6 Ψ(.),θ(.),p(observer) is really your ‘null’ model for evaluating factors influencing child-level Giardia infection probability (Ψ) and stool-level Giardia infection or shedding probability (ϑ). If I were interpreting these results, I would say there is strong evidence that Giardia infection probability (Ψ) varies with gender and is higher for girls (Ψ Girls = 0.54; SE(Ψ Girls) = 0.14) compared to boys (Ψ boys = 0.34; SE(Ψ boys) = 0.10; estimates based on the best-supported model, Figure 3). The gender effect is included in each of the top 4 models and 4 of the 5 models that perform better than the null model (Model 6, S1 Table). There is some evidence for an additive effect of age (β ^_age=0.26; SE(β ^_age)= 0.23) and social vulnerability (β ^_int=-1.00; SE(β ^_int)= 0.87; β ^_high=-0.20; SE(β ^_high)= 0.97) based on the best-supported model with these covariates, but the effects are imprecise. We do note that the direction of the effects is consistent with our expectations, with higher probabilities of Giardia infection probabilities for older children and those that reside in the Intermediate and High vulnerability sectors of the Federal District. The best-supported structure (hypothesis) for shredding probability suggests that this probability does not vary among sample order, gender, or age. The probability of stool-sample–level Giardia shedding, given child infection was (θ ^=0.44; SE(θ ^=0.12); estimates based on the best-supported model, S1 Table). There was some evidence that the shedding probability was slightly higher for older children (β ^_(theta-age)=0.31; SE(β ^_(theta-age))= 0.25) based on the second-best model, but these estimates are imprecise. This paragraph illustrates how you can use the model estimates together with the model selection table to interpret the result for your reader.

I know there is some model uncertainty in your candidate model set, but I don’t think the model-averaged estimates are that useful or necessary. I think you could include some of those estimates as an added column in Table 2 (if you chose to retain it) to illustrate the bias that exists using the raw proportions, but I don’t think Figure 4 is that useful or informative. A figure showing the relationship between child age (continuous variable) and shedding probability, based on the 2nd best-supported model would probably be more interesting.

Reviewer #4: This section, along with the tables and figures, is appropriate, precise, and of high quality, aiding in understanding the findings.

I would like to ask the authors why statistical tests to assess the significance of differences (such as P values) were not applied in Table 2. Was this outside the scope of the study, or were there specific technical considerations?

**Conclusions**

-Are the conclusions supported by the data presented?

-Are the limitations of analysis clearly described?

-Do the authors discuss how these data can be helpful to advance our understanding of the topic under study?

-Is public health relevance addressed?

Reviewer #1: yes

Reviewer #2: The conclusion are very reasonable. I like that authors have highlighted that certain molecular techniques might get around the issue of intermittent shedding.

Reviewer #3: In general, I believe the conclusions supported in the Discussion section reflect the data analysis well. I only have minor editorial comments on this section (see attached pdf).

Reviewer #4: As suggested in the journal guidelines, I prefer that the discussion begin with a summary of the key findings. Currently, the first paragraph of the discussion reads more like introductory material.

The results of the models in this paper are striking, if not shocking. For example, the estimated infection rate is almost double the observed infection rate. Likewise, the estimated infection rate for girls is approximately 1.6 times higher than for boys. Moreover, 70-80% of Giardia infections may go undetected with a one-sample/one-test strategy. Are these large figures reliable, or are there doubts about them? Is there any evidence from other populations to support these numbers? Is there a way to validate these estimates? Additionally, do existing studies corroborate the estimations, like gender differences in Giardia infection? These findings are significant enough to warrant further discussion on their reliability.

It might also be worthwhile for the authors to elaborate on how these findings contribute to the current body of knowledge.

The limitations are well-explained and realistic. As suggested by the journal, I would recommend that the authors also provide suggestions for future research in this section.

**Editorial and Data Presentation Modifications?**

Reviewer #1: accept

Reviewer #2: (No Response)

Reviewer #3: See attached pdf with minor recommendations.

Reviewer #4: The supplementary information is comprehensive and helpful, and I appreciate the authors’ efforts in providing it.

**Summary and General Comments**

Reviewer #1: This is an interesting statistical analysis and modelling of Giardia shedding and detection which coulld have implications on on diagnostic tests assessments and estimations of disease burden.

Minor suggestions for language revisions:

288: …somewhat higher= …higher 

291: small differences = trivial differences

295:…somewhat lower= …lower

390: rather inconsistently = inconsistently

391: perhaps increased: increased

416: understand= understanding

426: no support whatsoever= no support

429: take imperfect test sensitivity = take the imperfect test sensitivity

Reviewer #2: I think the paper makes a good contribution, and is really relevant to NTD community. If possible, it would be good to simplify the presentation of the modeling and results, because it is hard for people not familiar with this approach to understand. Having a statistical reviewer familiar with these models would be good.

Reviewer #3: The strength of this paper is the application of a hierarchical modeling approach to estimate Giardia infection in children attending kindergartens in the Federal District, Brazil while accounting (estimating) for stool-sample–level shedding rates and imperfect test-level detection. This same modelling approach could be used to separate lab-based diagnostics from disease related pathogen/parasite availability within field collected samples. The authors demonstrate this approach that could easily be used for other disease systems.

Reviewer #4: The manuscript presents an inspiring and insightful study on diagnostic tests and their limitations, using Giardia in children as a case study. It opens avenues for improving diagnostic sensitivity and provides a better understanding of the clinical and epidemiological characteristics of parasites. The study is well-designed, with information reported carefully, and while the work involves complex modeling with substantial calculations, the explanations are sufficient even for readers who may not be familiar with the subject. I commend the authors on their excellent work, and I hope this manuscript finds its deserved place in the scientific community. Here, I outline my points of confusion and suggestions for improvement. I would be happy to review a revised version and appreciate the authors' reasoning if they choose not to address certain suggestions.

Title:

The full title is important as it will represent the paper going forward. I believe it would be beneficial for the full title, like the short title, to specifically mention Giardia and modelling to better reflect the nature of the study. The current title doesn’t convey this adequately.

Abstract:

I think breaking down the sections of the abstract (Background, Methodology, Results, Conclusions) might help the reader gain a clearer understanding.

Author Summary:

If this section is aimed at a broader audience, particularly non-specialists, it would be useful to briefly explain the methodology, especially the hierarchical model used. Since the focus is more on the application of this model to estimate diagnostic success and its two key parameters, a short explanation of the model and its importance seems essential.

Introduction:

The introduction is well-written and structured, explaining fundamental issues in an accessible manner. However, the final paragraph could be written more clearly, as it is a bit confusing in how it lays out the two-level objectives of the study. Additionally, it might be beneficial to reference the background of the model used and how similar modeling approaches have previously been applied to other diseases, assisting researchers, clinicians, and policymakers.

There are also somewhat repetitive explanations regarding the model on page 10, lines 242-247, page 13, lines 309-315, and page 18, lines 419-422, which echo what is stated in the final paragraph of the introduction. The authors may consider consolidating these explanations in one section and providing only summaries or omitting them in others.

If this paper is examining something for the first time or has any other significant achievement, it would be helpful to briefly mention it at the end of the introduction.

I hope these suggestions and comments prove helpful, and I look forward to seeing the revised manuscript.

PLOS authors have the option to publish the peer review history of their article (what does this mean?). If published, this will include your full peer review and any attached files.

Reviewer #1: Yes: Ahmed A. Adeel

Reviewer #2: No

Reviewer #3: No

Reviewer #4: Yes: Amir-Hassan Bordbari
---

## [Decision Letter · Decision Letter 1]

23 Nov 2024

Dear Dr. Gurgel-Gonçalves,

We are pleased to inform you that your manuscript 'Disentangling the effects of intermittent faecal shedding and imperfect test sensitivity on the microscopy-based detection of gut parasites in stool samples' has been provisionally accepted for publication in PLOS Neglected Tropical Diseases.

Best regards,

Alberto Novaes Ramos Jr

Academic Editor

Guilherme Werneck

Section Editor

Shaden Kamhawi

co-Editor-in-Chief

Paul Brindley

co-Editor-in-Chief

Reviewer's Responses to Questions

**Key Review Criteria Required for Acceptance?**

**Methods**

-Are the objectives of the study clearly articulated with a clear testable hypothesis stated?

-Is the study design appropriate to address the stated objectives?

-Is the population clearly described and appropriate for the hypothesis being tested?

-Is the sample size sufficient to ensure adequate power to address the hypothesis being tested?

-Were correct statistical analysis used to support conclusions?

-Are there concerns about ethical or regulatory requirements being met?

Reviewer #1: Yes

Reviewer #2: All my comments have been adequately addressed

Reviewer #3: My previous review of this manuscript was complimentary. The authors apply a hierarchical modeling approach to estimate Giardia infection in children attending kindergartens in the Federal District, Brazil while accounting (estimating) for stool-sample–level shedding rates and imperfect test-level detection. This type of approach is appropriate, but rarely applied, to disentangling the effects of intermittent shedding and imperfect sensitivity and better understand transmission dynamics, disease epidemiology, and diagnostic-test performance.

The authors addressed many of my previous suggestions. I have a different philosophy to model building and model fitting than the authors, but I do not believe that their philosophy is necessary wrong or leads to erroneous scientific inference.

Reviewer #4: (No Response)

**Results**

-Does the analysis presented match the analysis plan?

-Are the results clearly and completely presented?

-Are the figures (Tables, Images) of sufficient quality for clarity?

Reviewer #1: yes

Reviewer #2: All my comments have been adequately addressed

Reviewer #3: The analysis presented does follow the design outlined by the authors. My previous review gave several suggestions on how to improve the clarity and depth of inference provided by their analysis.

The authors incorporated few of my suggestions, which I found this a little disappointing. Still, I acknowledge that I am used to writing and reviewing papers in the ecological literature where many of these hierarchical models were developed and where their use has become mainstream. Hopefully the author's paper will encourage adoption and application of these models in the NTD community. I appreciate the changes the authors did make and thank them for responding to my previous comments.

Reviewer #4: (No Response)

**Conclusions**

-Are the conclusions supported by the data presented?

-Are the limitations of analysis clearly described?

-Do the authors discuss how these data can be helpful to advance our understanding of the topic under study?

-Is public health relevance addressed?

Reviewer #1: yes

Reviewer #2: All my comments have been adequately addressed

Reviewer #3: Yes, I believe the conclusions supported in the Discussion section reflect the data analysis well. I have no further comments.

Reviewer #4: (No Response)

**Editorial and Data Presentation Modifications?**

Reviewer #1: Accept

Reviewer #2: All my comments have been adequately addressed

Reviewer #3: My previous review gave several editorial suggestions and recommendations on results presentation. The authors incorporated some of these suggestions. I have no new suggestions and I do not believe that repeating my previous suggestions, that were not incorporated, will likely lead to their adoption.

Reviewer #4: (No Response)

**Summary and General Comments**

Reviewer #1: Overall, this is a well-conceived and well implemented study and the results could stimulate further thinking about data of diagnostic tests and disease burden.

Reviewer #2: All my comments have been adequately addressed

Reviewer #3: The strength of this paper is the application of a hierarchical modeling approach to a case study to

estimate Giardia infection in children attending kindergartens in the Federal District, Brazil while

accounting (estimating) for stool-sample–level shedding rates and imperfect test-level detection.

The authors demonstrate this approach that could easily be used for other disease systems or to identify drivers of transmission in other Giardia systems and improving test performance.

Reviewer #4: I support the revised title: “Disentangling the effects of intermittent faecal shedding and imperfect test sensitivity on the microscopy-based detection of gut parasites in stool samples: Giardia as a case-study”.

PLOS authors have the option to publish the peer review history of their article (what does this mean?). If published, this will include your full peer review and any attached files.

Reviewer #1: **Yes: **Ahmed Adeel

Reviewer #2: No

Reviewer #3: No

Reviewer #4: No

---

## [Editor Report · Acceptance letter]

27 Nov 2024

Dear Dr. Gurgel-Gonçalves,

We are delighted to inform you that your manuscript, "Disentangling the effects of intermittent faecal shedding and imperfect test sensitivity on the microscopy-based detection of gut parasites in stool samples," has been formally accepted for publication in PLOS Neglected Tropical Diseases.

Best regards,

Shaden Kamhawi

co-Editor-in-Chief

Paul Brindley

co-Editor-in-Chief
